# Washable Fabric Triboelectric Nanogenerators for Potential Application in Face Masks

**DOI:** 10.3390/nano12183152

**Published:** 2022-09-11

**Authors:** Sang-Hwa Jeon, Yongju Lee, Swarup Biswas, Hyojeong Choi, Selim Han, Minseo Kim, Dong-Wook Lee, Sohee Lee, Hyeok Kim, Jin-Hyuk Bae

**Affiliations:** 1School of Electronic and Electrical Engineering, Kyungpook National University, Daegu 41566, Korea; 2School of Electrical and Computer Engineering, Center for Smart Sensor System of Seoul (CS4), University of Seoul, Seoul 02504, Korea; 3AI Robot R&D Department, Korea Institute of Industrial Technology (KITECH), Ansan 15588, Korea; 4School of Chemical Engineering, Pusan National University, Busan 56241, Korea; 5Department of Clothing and Textiles, Research Institute of Natural Science, Gyeongsang National University, Jinju 52828, Korea; 6School of Electronics Engineering, Kyungpook National University, Daegu 41566, Korea

**Keywords:** washability, fabric-triboelectric nanogenerators, degradation ratio, elastomer, flash-spun nonwoven

## Abstract

In order to counteract the COVID-19 pandemic by wearing face masks, we examine washable fabric-based triboelectric nanogenerators (FTENGs). We applied the flash-spun nonwoven fabric (FS fabric) into the FTENGs, comparing the melt-blown nonwoven fabric (MB fabric) based FTENGs, which is conventionally studied in the field of energy harvesting. For reusability, all our proposed FTENGs are systematically investigated by controlling the washing conditions. After washing, the degradation ratio of the obtained output voltage is found to be only 12.5% for FS FTENGs, compared to the ratio of about 50% for the typical MB FTENGs. A rather small degradation ratio for FS fabric cases has resulted from less changed fabric structure after washing due to more dense fabric nature. Additionally, in order to improve the electrical characteristics of FS FTENGs. Note that the output voltage of FTENGs exhibits as much as 600 V.

## 1. Introduction

Triboelectric nanogenerators (TENGs) have attracted increasing attention over the past 10 years as self-powered sensors or eco-friendly energy harvesters [1,2,3,4,5,6,7,8]. The fundamental concept behind TENGs is the conversion of irregular motion energy, which is generally wasted, into useful electrical energy based on contact electrification and electrostatic induction phenomena [9]. TENGs can be fabricated using a variety of materials [10]; however, fabric-based TENGs (FTENGs) are particularly promising as wearable electronic devices because they are flexible, stretchable, and produced from materials that are similar to those used in existing clothing [11,12,13,14,15,16]. Meanwhile, the recent COVID-19 outbreak has necessitated the use of masks. Apart from COVID-19, growing concerns over air pollution and respiratory health also contribute to the demand for masks. This has motivated researchers to actively improve the functionality and reusability of masks. Additionally, they have also received attention as form factors in the wearable electronics field [17,18,19]. Given that several people currently wear quarantine masks made of fabric, FTENGs demonstrate significant potential as energy sources or sensors for wearable electronic devices. In fact, following the COVID-19 pandemic, triboelectric nanogenerator face mask research is currently underway [20,21,22].

However, mass-produced, melt-blown nonwoven-filter quarantine masks that are currently in use typically generate a considerable amount of waste, because they are disposable, which adversely affects the environment [23]. Therefore, growing interest has been noted in the “reusability” of quarantine masks. Moreover, considering their applications as wearable electronic devices, reusability may be one of the more important factors. Reusable clothing must be washable, and the existing characteristics should not change significantly as a result of washing. The efficiency of wearable electronic devices could also deteriorate over time if their characteristics change significantly owing to activities such as washing.

With these considerations, in this study, we investigated the use of fabric materials suitable to produce reusable FTENGs, including their ability to endure stressful conditions such as washing. The melt-blown nonwoven fabric, which is primarily used in existing quarantine masks, and the flash-spun nonwoven material, proposed in this study, were both used to produce the FTENGs. In addition, we fabricated a FTENGs with an electrode-inserted structure (+: positive charge, −: negative charge) to improve the performance of the FTENGs (Figure 1a,b). This structure induces more charge per unit area and improves the output characteristics.

This enabled us to assess the changes in the electrical characteristics and physical properties of such devices under various washing conditions, including the temperature and washing time. In the case of the melt-blown nonwoven fabric, which is used in existing quarantine masks, the electrical characteristics changed significantly after washing; however, the characteristics of the proposed flash-spun non-woven fabric were relatively stable. In addition, a polydimethylsiloxane (PDMS) layer was introduced to enhance the output characteristics of the FTENGs fabricated using the flash-spun nonwoven material. The mere introduction of this additional layer increased the output voltage by as much as five times.

## 2. Materials and Methods

### 2.1. Fabrication of Fabric-Based TENGs

Figure 2a illustrates the structure of the contact-mode FTENGs fabricated in this study. Each FTENG was manufactured by inserting a copper tape as an electrode between two fabric sheets with dimensions of 4 × 4 cm^2^. Both melt-blown nonwoven fabric (MB fabric) (Korea Institute of Industrial Technology, Ansan, Republic of Korea) and flash-spun nonwoven fabric (FS fabric) (Tyvek, Dupont, DE, USA) were used in this study. Figure 2b illustrates photographic images of sheets of the MB and FS fabric, which consist of polypropylene and high-density polyethylene fibers, respectively. To enhance the output of the FS FTENGs, a PDMS layer containing an electrode was introduced into the FTENGs. The PDMS insert was prepared as follows: first, a PDMS elastomer solution was spin-coated onto a silicon wafer at 300 rpm and annealed at 80 °C. Then, silver was deposited onto the PDMS coating using an evaporator to form a silver electrode, an additional elastomer solution was spin-coated onto the silver layer, followed by annealing under the same conditions. Sylgard-184, an elastomeric PDMS kit from Dow Corning, was used to produce the PDMS elastomer.

### 2.2. Washing and Measurement of the Characteristics of the FTENGs

Figure 2c illustrates a schematic of a washing experiment. The effect of washing conditions on the FTENGs was investigated by washing the MB and FS fabrics with water and detergent (AATCC 1933 Standard Reference Detergent WOB (AATCC, Durham, NC, USA)) at various temperatures (40, 50, and 60 °C) and for different periods of time (20, 30, and 40 min). The thickness of each sheet of fabric was measured before and after washing. Field emission scanning electron microscopy (FE-SEM) (S-4800, Hitachi High-Technology, Tokyo, Japan) was also used to observe the microstructural changes in the fabric. An oscilloscope (MDO3052, Tektronix, OR, USA) was used to measure the output voltages of the FTENGs fabricated in this study at various pressure levels.

## 3. Results and Discussion

As mentioned above, to apply FTENGs as a reusable device, a fabric with little change under stressful conditions such as washing is essential. Figure 3 illustrates the variation in the thicknesses of the MB and FS fabric sheets with the number of washings, washing temperatures, and washing times. The average values and standard deviations were extracted by measuring the thicknesses of the MB and FS fabric sheets 20 times under each condition. The thin linear error bars in Figure 3d represent the standard deviation, and the points in Figure 3 represent the average thickness. For the sheets of both fabrics, it was challenging to determine the tendency of the thickness variation in response to the washing conditions. Even though the change in the thickness of the fabric sheet after washing under different conditions did not exhibit a clear tendency, it is evident from the graph that the variations in the thicknesses of the MB fabric sheets were much greater than those of the FS fabric sheets. After one wash, the standard deviation of the MB fabric sheet varied from a minimum of 0.023 mm (60 °C, 20 min) to a maximum of 0.06 mm (50 °C, 30 min). After two washes, the minimum and maximum thicknesses were 0.006 mm (60 °C, 40 min) and 0.031 mm (40 °C, 40 min), respectively. In comparison, after one wash, the standard deviation of the FS fabric sheet varied between 0.002 mm (60 °C, 20 min) and 0.015 mm (40 °C, 30 min), and, after two washes, the minimum and maximum thicknesses were 0.001 mm (40 °C, 20 min) and 0.014 mm (40 °C, 30 min), respectively. Overall, the deviation in the thickness of the FS fabric sheet was much smaller than that of the MB fabric sheet. Bearing in mind that the electrical performance of a FTENG with a layered structure may be affected by the layer thickness [24]; a smaller variation in the thickness of the FS sheet may contribute to the stability of the electrical characteristics of the FTENGs.

Figure 4 illustrates FE-SEM images of the surfaces of the MB (Figure 4a,b) and FS (Figure 4c,d) fabrics, respectively. As their names suggest, MB fabrics and FS fabrics are produced by different manufacturing processes; hence, the two fabrics exhibit significantly different microstructures, namely single-fiber and networked fiber morphologies. Contrary to the MB fabric, the FS fabric comprises individual fibers that are wide and flat, with little space between the fibers. In addition, the connectivity between the fibers seems to be high. The microstructure of the MB fabric appears to be severely degraded after washing, as illustrated in Figure 4b, with a significant increase in the amount of empty space between individual fibers, and the fibers also appear to be distributed unevenly and irregularly compared with the unwashed fabric (Figure 4a). Alternatively, the FS fabric (Figure 4c,d) is degraded to a much lesser extent during washing. Compared with its appearance before washing, gaps can be observed to appear between fibers, and the connection is weakened slightly; however, the degree of degradation is small compared with that in the MB fabric, and the distribution of fibers remains uniform. The results in Figure 4 indicate that the microstructure of the MB fabric deteriorates significantly during the washing process compared with that of the FS fabric. This implies that the FTENGs produced using the MB fabric are more vulnerable to stressful conditions, such as washing, than FTENGs produced using the FS fabric. Additionally, the wide variability of the thickness of the MB fabric sheet as a result of washing, as illustrated in Figure 3, corresponds well with these FE-SEM results.

Figure 5 illustrates the variations in the output voltage of the MB and FS FTENGs according to the washing temperature and the number of times. The output voltage was measured at a temperature of 25 °C, humidity of 65%, and pressure of 22 N in air. The variation in the output voltage of both the FTENGs after 20 min of washing at various temperatures is illustrated in Figure 5a (MB-40 °C and MB-50 °C, FS-40 °C and FS-60 °Care overlapped because their values are very similar). The output voltage decreased by at least 45% (at 60 °C) and by as much as 60% (at 40 °C) in the case of the MB FTENG. For the FS FTENG, the output voltage increased slightly (at 50 °C) and decreased by as much as 18.6% (at 60 °C) after washing. The variation in the output voltage of the FTENGs after 30 min of washing is illustrated in Figure 5b. In this case, the output voltage of the FTENG based on the MB fabric decreased by at least 50% (at 60 °C) and by as much as 58% (at 40 °C). In the case of the FTENG based on the FS fabric, the output voltage decreased by at least 17.9% (at 50 °C) and by as much as 56.8% (at 60 °C) after washing. Figure 5c depicts the variation in the output voltage of the devices after 40 min of washing with the configurations illustrated in Figure 5a,b. The output voltage of the FTENG based on the MB fabric decreased by at least 41.7% (at 60 °C) and by as much as 78.3% (at 40 °C) as a result of washing. The output voltage of the FTENG based on the FS fabric decreased by at least 18.6% (at 60 °C) and by as much as 55.7% (at 40 °C) after washing. Thus, the FTENGs based on the MB fabric exhibit a large decrease in the output voltage at a washing temperature of 40 °C and a small decrease at 60 °C. However, the variation tendency of the output voltage with respect to the washing temperature and time is not apparent from Figure 5, similar to the variation in the thickness of the fabric sheet in response to the washing conditions, as illustrated in Figure 3. Despite the high output voltage of the FTENG based on the MB fabric, the reduction in the output voltage was greater than that of the FS fabric under almost all washing conditions.

Table 1 summarizes the degradation ratio of output voltages measured as a function of the washing temperature and time. The degradation ratio is the ratio of the output voltage change by washing divided by the initial output voltage. The degradation ratio is an absolute value. For the FTENG based on the MB fabric, the output voltage decreased after washing. For FTENGs with MB fabric, it is difficult to predict the output after exposure to stressful processing such as washing, which impedes their application as washable devices. Alternatively, the variation in the output voltage of the FTENGs with FS fabric after washing was small, and the characteristics were more stable after washing compared with those of the FTENGs produced using the MB fabric. This result suggests that the FS fabric is more suitable for washable wearable devices than the MB fabric.

Figure 6a–d illustrates the output voltage before and after washing the FTENGs based on the MB and FS fabrics. The corresponding measurements were conducted at a pressure of 23 N and a frequency of 4 Hz. Figure 6e compares the maximum peak-to-peak voltage (VPP) obtained from the graph of the output voltage depicted in Figure 6a–d. The VPP values of the FTENGs consisting of other materials such as thermoplastic vulcanizates (TPVs) and PDMS are also illustrated in the Figure 6 for comparison. It is known that PDMS is strongly negatively charged by friction and is one of the most frequently investigated materials in the field of TENGs [25]. The polymer material known as TPV, which has recently gained recognition as a rubber-like material, is characterized by a structure, wherein cross-linked rubber particles are evenly dispersed in a thermoplastic resin [26]. In Figure 6e, the VPP of the FTENG based on the MB fabric decreases by approximately 43.8% after washing. Alternatively, the VPP of the FTENG based on the FS fabric decreases by approximately 12.5% after washing. Figure 6f illustrates VPP measured at a relatively low pressure of 2.2 N and a frequency of 6 Hz. In Figure 6f, the VPP of the FTENG based on the MB fabric decreases by approximately 41.2% from 10.2 V before washing to 6 V after washing. In contrast, the VPP of the FTENG based on the FS fabric decreases by approximately 18.9% from 3.5 V before washing to 2.84 V after washing. Even at a low pressure, the VPP of the FTENG based on the MB fabric decreased dramatically before and after washing, whereas the VPP of the FTENG based on the FS fabric was relatively stable.

The FTENG based on the FS fabric clearly demonstrates good stability with respect to washing; however, it presents a disadvantage in that its output voltage is relatively low compared with that of the FTENG based on the MB fabric. By introducing an additional PDMS layer with an inserted electrode, the output voltage of the FTENG based on the FS fabric was enhanced to compensate for this weakness. Figure 7a–c illustrates the schematics of the FTENGs, wherein the PDMS layer is introduced at different positions in the stack. Figure 7d–f illustrate the output voltage corresponding to each of the FTENGs illustrated in Figure 7a–c. The output voltages of all the FTENGs were measured at a temperature of 20.6 °C, humidity of 61%, and pressure of 28 N in air. The output voltage of the FTENG with the PDMS layer placed on top of the fabric is plotted in Figure 7d. The VPP of this FTENG is 672 V. Figure 7e,f illustrate the output voltages of the FTENGs with the PDMS layer placed between the two fabric layers and below the fabric layers, respectively. The VPP in the former case (Figure 7e) is 618 V, and that in the latter case (Figure 6f) is 250 V. The simple method that entailed the introduction of the PDMS layer increased the VPP by nearly five times compared with the VPP of the FTENG based on the FS fabric (Figure 6). The reason was that, when an additional layer was inserted, it became double electrode mode, and thus a voltage was compensated for, and the negative charge affinity of PDMS in the triboelectric series was greater than that of the fabric, so more voltage was generated.

## 4. Conclusions

To summarize, this study led us to propose washable FTENGs that can be used to develop mask-form wearable devices. The MB fabric, which is primarily utilized in existing quarantine masks, and the FS fabric were selected as fabrics for manufacturing the washable FTENGs. The changes in the thicknesses of the fabric sheets were compared by varying the washing conditions (washing temperature and time). The change in the microstructure of each fabric after washing was confirmed through FE-SEM analysis. The microstructure of the FS fabric underwent a significantly smaller change than that of the MB fabric, based on the thickness variation and the FE-SEM images. Although the FTENG based on the MB fabric was generally found to produce a high output voltage, the output voltage of the FTENG based on the MB fabric decreased by 43.8% after washing, whereas that of the FTENG based on the FS fabric decreased by 12.5%. These results suggest that the FS fabric is a more suitable material than the MB fabric when incorporated into washable FTENGs. Additionally, a PDMS layer with an inserted electrode increased the output voltage of the TENGs by nearly fivefold to compensate for the relatively low output voltage of the FTENG based on the FS fabric. We intend to conduct an experiment on a face mask using the developed fabric-based TENG device in the future

## Figures and Tables

**Figure 1 nanomaterials-12-03152-f001:**
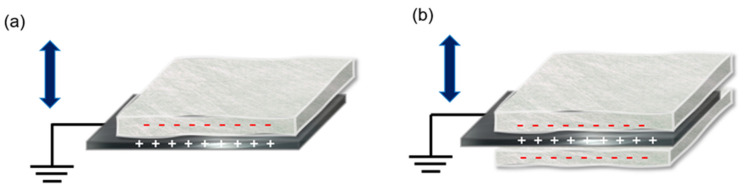
Working principle of (**a**) original FTENG and (**b**) electrode insertion type FTENG. (+: positive charge, −: negative charge.

**Figure 2 nanomaterials-12-03152-f002:**
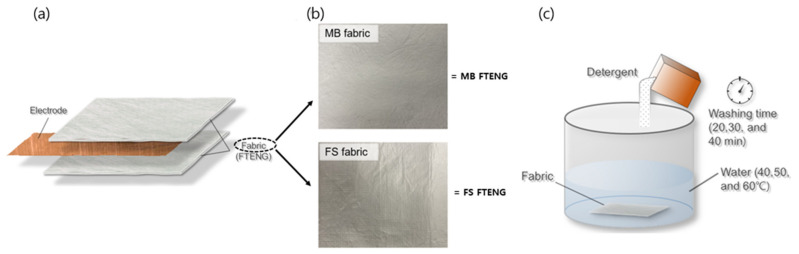
(**a**) Schematic of a FTENG. Photographic surface images of (**b**) melt-blown nonwoven (MB) and flash-spun nonwoven (FS) fabric sheets and (**c**) experimental schematic of washing conditions.

**Figure 3 nanomaterials-12-03152-f003:**
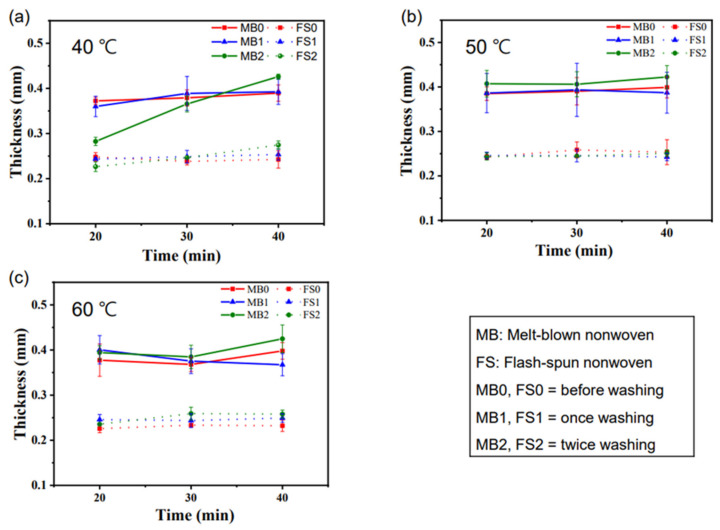
Variations in the fabric thickness vs. the washing time and number of washes. The washing temperatures are (**a**) 40, (**b**) 50, and (**c**) 60 °C, respectively.

**Figure 4 nanomaterials-12-03152-f004:**
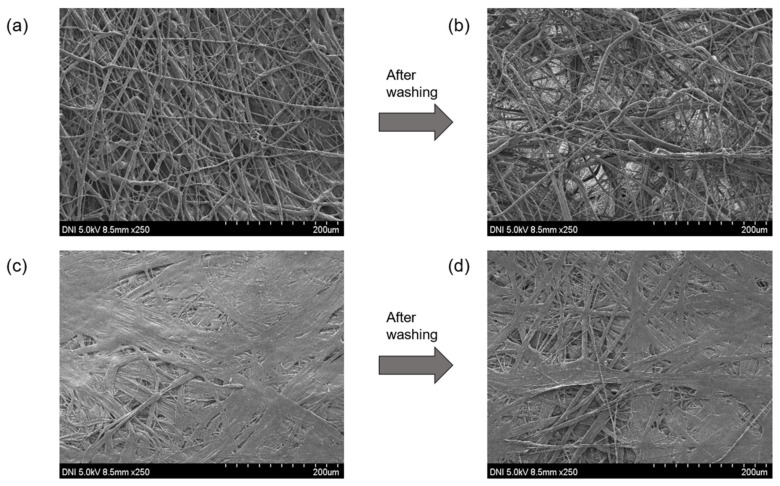
FE-SEM images of the MB nonwoven fabric surface (**a**) before and (**b**) after washing and of the FS nonwoven fabric surface (**c**) before and (**d**) after washing.

**Figure 5 nanomaterials-12-03152-f005:**
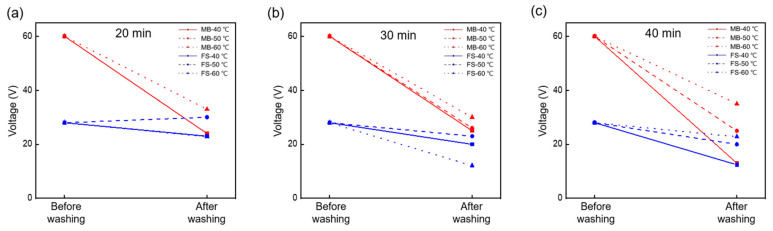
Variation in the maximum peak-to-peak voltage of the FTENGs based on the FS and MB fabrics after washing for (**a**) 20, (**b**) 30, and (**c**) 40 min. The red and blue lines represent the voltages of the FTENGs based on the MB and FS fabric, respectively. The solid, dashed, and dotted lines represent washing at 40, 50, and 60 °C, respectively.

**Figure 6 nanomaterials-12-03152-f006:**
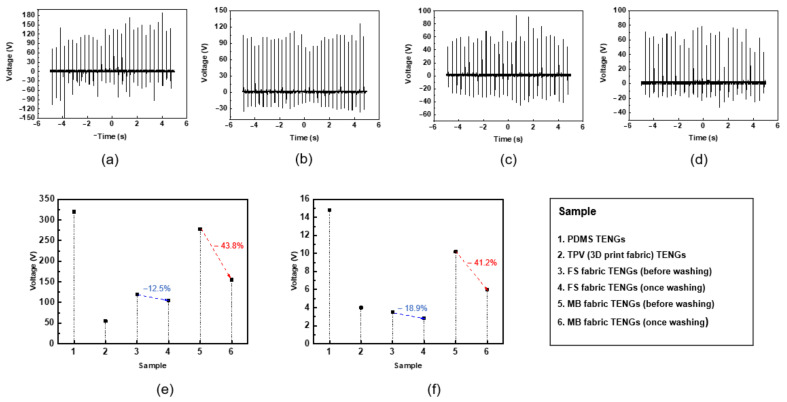
Output voltage of the FTENGs based on the MB fabric (**a**) before and (**b**) after washing and of the FTENGs based on the FS fabric (**c**) before and (**d**) after washing at high pressure (23 N). (**e**) Maximum peak-to-peak voltage of the FTENGs based on different fabric sheets at (**e**) high and (**f**) low pressure (2.2 N).

**Figure 7 nanomaterials-12-03152-f007:**
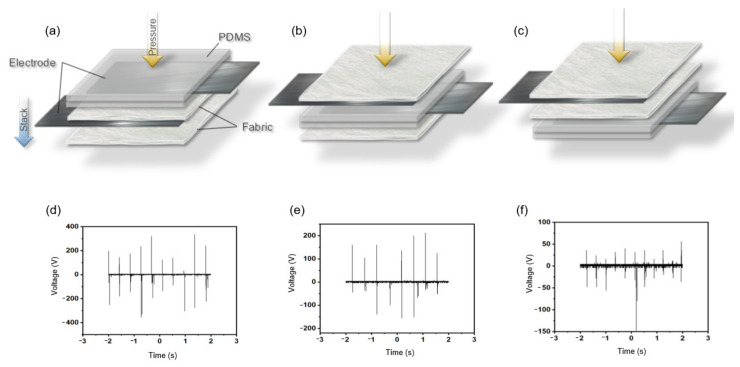
Schematic of the FTENGs with the PDMS layer positioned (**a**) at the top, (b) in the middle, and (**c**) at the bottom. Output voltage of the FTENGs with the PDMS layer (**d**) at the top, (**e**) in the middle, and (**f**) at the bottom.

**Table 1 nanomaterials-12-03152-t001:** The degradation ratio of output voltages of the FTENGs based on the FS fabric and the MB fabric under various washing conditions.

**MB** **FTENGs**	Time (min)	**20**			**30**			**40**		
	Temperature (°C)	40	50	60	40	50	60	40	50	60
	Degradation Ratio (%)	60	60	45	58.3	56.7	50	78.3	58.3	41.7
**FS** **FTENGs**	Time (min)	**20**			**30**			**40**		
	Temperature (°C)	40	50	60	40	50	60	40	50	60
	Degradation Ratio (%)	17.9	7.1	18.6	28.6	17.9	56.8	55.7	28.6	18.6

## Data Availability

The data is available on reasonable request from the corresponding author.

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
