# Peer review of "Washable Fabric Triboelectric Nanogenerators for Potential Application in Face Masks"

_nanomaterials, 2022, doi:10.3390/nano12183152_

Round 1
Reviewer 1 Report
This work reported washable fabric-based triboelectric nanogenerators (FTENGs) and their ability to endure stressful conditions such as washing. The masks made of FTENG exhibit significant potential as energy sources or sensors for wearable electronic devices. It is suggested that this work may be accepted for publication after major revision by referring to the following comments.
1 Page 2 line 59, the word ‘FTNEGs’ should be corrected as FTENGs.
2 It is difficult to distinguish the results (points in Figure 2 a-b) of MB0, MB1 and MB2. Please modified the figures to make it easier for understanding.
3 In Figure 3a, the points of FS-60 and MB-50 are missing, please add the related results.
4 In order to highlight the advantages of this work, some recent important works of TENG should be referred. Such as Adv. Mater. 2022, 34, 2202238, DOI: 10.1002/adma.202202238; J. Mater. Chem. A, 2019, 7, 19485, DOI: 10.1039/c9ta06525c; and Sensors 2021, 21, 7129, DOI: 10.3390/s21217129.
5 The introduction of the PDMS layer increased the VPP by nearly five times compared with the VPP of the FTENG based on the FS fabric. The authors should add more explanations about this phenomenon.
Reviewer 2 Report
The authors introduced the washable fabric-based triboelectric nanogenerators (FTENGs). The authors applied the flash-spun nonwoven fabric into the FTENGs, and explored their performance before and after washing. The degradation ratio of the obtained output voltage remained 12.5% for FS FTENGs. We would like to suggest accepting the article after a major revision.
1. The authors claim in the title, introduction and conclusion that this work is used for face mask in the application on COVID-19 pandemic. But this article almost not mention how the device can used as a mask. Further explorations are required, including: 1) output, the filtering ability of nanoparticles, the antibacterial performance of the FTENG used as face mask. And the breathability of the FTENG.
2. The working principle of FTENG are suggested to add in the main Figures.
3. We are confused about the difference among MB TENG, FS TENG and FTENG, could the authors provide a clearer schematic.
4. There are some text mistakes in the manuscript, like, (Line 158-159: Figure 4 illustrates the variations in the output voltage of the MB and FS FTENGs according to the washing temperature and.) This sentence seems incomplete.

Round 2
Reviewer 1 Report
The related issues have been addressed, and the revised version of the manuscript meets the criteria for publication in 'Nanomaterials'.Reviewer 2 Report
The authors have full covered our questions.